# Alzheimer’s Amyloid β Peptide Induces Angiogenesis in an Alzheimer’s Disease Model Mouse through Placental Growth Factor and Angiopoietin 2 Expressions

**DOI:** 10.3390/ijms24054510

**Published:** 2023-02-24

**Authors:** Abdullah Md. Sheikh, Shozo Yano, Shatera Tabassum, Shingo Mitaki, Makoto Michikawa, Atsushi Nagai

**Affiliations:** 1Department of Laboratory Medicine, Shimane University School of Medicine, 89-1 Enya Cho, Izumo 693-8501, Japan; 2Department of Neurology, Shimane University School of Medicine, 89-1 Enya Cho, Izumo 693-8501, Japan; 3Department of Biochemistry, Graduate School of Medical Sciences, Nagoya City University, Nagoya 467-8601, Japan

**Keywords:** Alzheimer’s disease, amyloid β peptide, angiogenesis, placental growth factor, angiopoietin 2

## Abstract

Increased angiogenesis, especially the pathological type, has been documented in Alzheimer’s disease (AD) brains, and it is considered to be activated due to a vascular dysfunction-mediated hypoxic condition. To understand the role of the amyloid β (Aβ) peptide in angiogenesis, we analyzed its effects on the brains of young APP transgenic AD model mice. Immunostaining results revealed that Aβ was mainly localized intracellularly, with very few immunopositive vessels, and there was no extracellular deposition at this age. Solanum tuberosum lectin staining demonstrated that compared to their wild-type littermates, the vessel number was only increased in the cortex of J20 mice. CD105 staining also showed an increased number of new vessels in the cortex, some of which were partially positive for collagen4. Real-time PCR results demonstrated that placental growth factor (PlGF) and angiopoietin 2 (AngII) mRNA were increased in both the cortex and hippocampus of J20 mice compared to their wild-type littermates. However, vascular endothelial growth factor (VEGF) mRNA did not change. Immunofluorescence staining confirmed the increased expression of PlGF and AngII in the cortex of the J20 mice. Neuronal cells were positive for PlGF and AngII. Treatment of a neural stem cell line (NMW7) with synthetic Aβ_1–42_ directly increased the expression of PlGF and AngII, at mRNA levels, and AngII at protein levels. Thus, these pilot data indicate that pathological angiogenesis exists in AD brains due to the direct effects of early Aβ accumulation, suggesting that the Aβ peptide regulates angiogenesis through PlGF and AngII expression.

## 1. Introduction

Alzheimer’s disease (AD) is a common dementia disease characterized by a progressive decline in cognitive functions [1]. Pathologically, amyloid plaques and intraneuronal neurofibrillary tangles are the main diagnostic criteria of AD [2]. Amyloid plaques primarily contain an aggregated form of amyloid β (Aβ), a 39–42 amino acids-long peptide fragment generated from membranous amyloid precursor protein (APP) by β- and γ-secretase enzyme activities [2,3]. This peptide is aggregation-prone and deposited in the brain parenchyma as oligomers or amyloid fibrils [4]. Aggregated Aβ shows neurodegenerative and neuroinflammatory properties [5,6]. These features (neurodegeneration and neuroinflammation) are always found in AD brains, indicating the potential importance of aggregation and deposition of the Aβ peptide in the pathology [3]. Probable causes of Aβ deposition are suggested to be increased production or decreased clearance [7]. After production, Aβ is cleared from the brain by enzymatic degradation and by phagocytic cells [8,9,10,11]. However, a bulk of the peptide is cleared through perivascular pathways [7]. Due to its high aggregation properties, Aβ may aggregate during its clearance and interfere with the perivascular pathways. Hence, the deposition of the peptide around the vessels could be an important feature of AD pathology. Indeed, deposition of Aβ is frequently seen around the vessels that cause amyloid angiopathy, along with AD pathology [12]. Such deposition causes the vessel to become fragile, resulting in microbleeds and associated inflammation [13].

In addition to neurodegeneration and neuroinflammation, increased angiogenesis is frequently seen in AD brains [14]. Aβ peptide demonstrated toxicity, not only towards neurons, but also towards the vascular cells, including endothelial cells and smooth muscle cells [15,16]. It is suggested that Aβ deposition might cause the dysfunction of vessel function, resulting in a hypoxic condition in AD brains [17]. Additionally, vascular dysfunction can induce an inflammatory response [18]. All these signals trigger the angiogenic process [14,19]. Hence, angiogenesis in AD is considered an indirect consequence of Aβ deposition-dependent vascular dysfunction. Such angiogenesis is mainly a pathologic type, which might cause the extravasation of blood constituents and the aggravation of the neuroinflammatory condition [10]. However, Aβ can directly induce inflammatory conditions [20]. Since inflammation and angiogenesis are intimately related [21], it is possible that Aβ can directly induce angiogenesis long before the occurrence of vascular dysfunctions and the related inflammation. Therefore, we hypothesized that angiogenesis is an early feature of AD pathology, which is a direct consequence of excess Aβ in the brain.

In this study, we investigated the direct effects of Aβ on angiogenesis in AD using disease model mice and in vitro cell culture systems. To eliminate the role of vascular dysfunction in AD angiogenesis, we used young AD model animals prior to Aβ deposition in the vessels. We found that Aβ can directly induce pathological angiogenesis by altering the expression of several angiogenesis factors.

## 2. Results

### 2.1. Aβ Deposition and Vessel Density in J20 Mice Brains

Increased angiogenesis has been documented in AD brains due to vascular dysfunction and hypoxia [22,23]. Since Aβ deposition in vessel walls is considered a main pathological cause of vascular dysfunction and subsequent angiogenesis in AD [3,17], it was evaluated in APP transgenic mice (J20 strain) brains at an earlier time point (2 months). Immunostaining results showed that Aβ was mainly intraneuronal in the cortex and hippocampal areas at this time point, and very few vessels were Aβ immunopositive (Figure 1A and Appendix A). The staining demonstrated a wide distribution pattern of Aβ in the cortex, but in the hippocampus CA1 areas, Aβ mainly positive in the pyramidal cell layer (Figure 1A). Quantification of the staining showed that at 2 months of age, the levels of Aβ in the J20 mice brains were higher than those in the wild-type mice (cortex Wt: 0.1 ± 0.09 vs. J20: 2.9 ± 1.2, *p* < 0.001; hippocampus Wt: 0.1 ± 0.09 vs. J20: 2.4 ± 0.75, *p* < 0.001), in which they was almost negative (Figure 1B). Then, double immunofluorescence staining was performed to identify Aβ-positive cells. The results demonstrated that Aβ was mainly positive in NeuN-positive neurons at this age (Appendix A). Since the antibody used for Aβ staining (6E10) can also detect amyloid precursor protein (APP) [24], immunostaining was performed using an APP-specific antibody. The results showed that at this time point, the staining pattern of APP was comparable to Aβ (Appendix A). Moreover, APP-positive areas in the cortical and hippocampal areas of wild-type mice were similar to those of J20 (Appendix A). At 15 months of age, extracellular deposits of Aβ were found both in the cortical and hippocampal areas of J20 mice, and the immunopositive areas in both regions were increased compared to those of the 2-month-old mice (Appendix A). Since Aβ has the propensity to oligomerize, immunostaining was performed using an oligomer-specific antibody. The results showed that anti-oligomer immunopositive cells were mainly located in the cortical areas, with a few positive cells in the hippocampus (Figure 1C). Quantification of the staining revealed that the anti-oligomer immunopositive areas were increased both in the cortex and hippocampus of the J20 mice compared to the wild-type mice, in which they were almost negative (cortex Wt 0.19 ± 0.06 vs. J20 3 ± 0.76, *p* < 0.01; hippocampus Wt 0.22 ± 0.05 vs. J20 0.53 ± 0.17, *p* < 0.05) (Figure 1C,D). The vessel staining with STL showed that the number was increased only in the cortex (Wt: 30.5 ± 1.4 vs. J20: 43.7 ± 1.7, *p* < 0.001), but not in the hippocampal areas of J20 mice (Wt: 22.5 ± 2.1 vs. J20: 27.1 ± 4.5, *p* = 0.08) (Figure 1E,F).

### 2.2. Evaluation of Angiogenesis in J20 Brains

Endothelial cells, especially angiogenic endothelial cells, express CD105 [25]. Immunostaining of CD105 showed a round-shaped appearance, along with long vessel-like structures in the cortex of the mice. The areas of such round-shaped CD105 positive structures were increased in J20 mice at 2 months of age (Wt: 2.4 ± 0.38 vs. J20: 6 ± 0.59, *p* < 0.05) (Figure 2A,B). However, in the hippocampus, the CD105-positive areas appeared to be similar between the Wt and J20 mice (Figure 2A,B).

Next, the types of newly formed vessels were evaluated by double immunofluorescence staining of CD105 and collagen4 (Col4), where Col4 was used as a basement membrane marker. The results demonstrated that in the cortex of the J20 mice brains, many CD105-positive vessels lacked Col4 (Figure 2C).

### 2.3. Evaluation of the Expression of Angiogenesis Regulators in J20 Mice Brains

First, the expression of angiogenesis regulators was evaluated at mRNA levels. The real-time PCR results showed that the mRNA of angiogenesis inducers such as VEGF was not increased in the cortex or hippocampal areas of J20 mice brains (cortex Wt: 0.88 ± 0.1 vs. J20: 0.81 ± 0.6, *p* = 0.044; hippocampus Wt: 0.74 ± 0.26 vs. J20: 1.32 ± 0.99, *p* = 0.23) (Figure 3A,B). However, the mRNA of PlGF, an angiogenesis inducer of the VEGF family [26], was increased in both the cortex and the hippocampus of J20 mice brains compared to those of their wild-type counterparts (cortex Wt: 1.17 ± 0.25 vs. J20: 2.94 ± 0.5, *p* < 0.01; hippocampus Wt: 1.12 ± 0.1 vs. J20: 5.2 ± 2.1, *p* < 0.05) (Figure 3A,B). Additionally, the mRNA of angiopoietin2 (Ang2), an angiogenesis regulator that destabilizes the vessels, was increased in those areas (cortex Wt: 1.47 ± 0.42 vs. J20: 3 ± 0.62, *p* < 0.05; hippocampus Wt: 2.19 ± 0.62 vs. J20 2.97 ± 0.15, *p* < 0.05) (Figure 3A,B). Conversely, the mRNA of Ang1 was not changed (cortex Wt: 0.87 ± 0.11 vs. J20: 0.78 ± 0.15, *p* = 0.21; hippocampus Wt: 1.06 ± 0.36 vs. J20: 1.61 ± 0.42, *p* = 0.08).

Then, the expression of angiogenesis regulators in J20 mice brains at the protein level was evaluated. The immunostaining results showed that although the % immunopositive area of VEGF protein was increased in the cortical areas, it was not changed in the hippocampus of J20 mice brains compared to their wild-type counterparts (cortex Wt: 0.52 ± 0.17 vs. J20: 2.5 ± 0.66, *p* < 0.01; hippocampus Wt: 0.5 ± 0.6 vs. J20: 0.74 ± 0.23, *p* = 0.11) (Figure 4A,C and Appendix A). Moreover, the levels of VEGF and HIF-1α were very low in both the cortex and hippocampus of Wt and J20 mice at this time point (Figure 4B,C and Appendix A).

Next, we evaluated the expression of PlGF and AngII in the J20 mouse brains at protein levels. Immunostaining results demonstrated that PlGF was expressed mainly in the neuron-like cells in the cortex (Figure 4D). Its expression in the hippocampus was lower than in the cortex. Quantification of immunostaining results showed that the % immunopositive area of PlGF protein was significantly increased in the cortex of J20 mice compared to their wild-type counterparts, whereas it was not changed in the hippocampus (cortex Wt: 1.36 ± 0.29 vs. J20: 4.84 ± 0.82, *p* < 0.01; hippocampus Wt: 2.05 ± 0.35 vs. J20: 1.56 ± 0.27, *p* = 0.39) (Figure 4F).

In the case of Ang2, the staining pattern showed a round-shaped appearance (Figure 4E). Similar to PlGF, the % immunopositive area of Ang2 protein was higher in the cortex compared to the hippocampus. Compared to the wild-type, Ang2 was increased both in the cortex and hippocampus of J20 mice (cortex Wt: 0.52 ± 0.21 vs. J20: 4.2 ± 0.98, *p* < 0.01; hippocampus Wt: 0.07 ± 0.05 vs. J20: 1.8 ± 0.3.1, *p* < 0.01) (Figure 4F).

### 2.4. Identification of PlGF and AngII Expressing Cells in J20 Mice Brain

To identify whether neurons expressed PlGF and AngII in J20 mouse brains, double immunofluorescence experiments were performed using NeuN neuronal labeling. The results showed that in J20 mice brains, many neurons were positive for PlGF in the cortex (Figure 5A and Appendix A). Some of the neurons in the hippocampus were also positive (Figure 5A). Conversely, a few neurons were positive for AngII only in the cortical areas (Figure 5B and Appendix A).

### 2.5. Effects of Aβ Peptide on PlGF and AngII Expression in an In Vitro Neural Stem Cell Culture

Since neurons expressed both PlGF and AngII in vivo in J20 mice brains, we investigated the direct effects of the Aβ peptide on the expression of PlGF and AngII in a mouse neural stem cell line (NMW7) culture. After stimulating NMW7 with a synthetic Aβ_1–42_ peptide, the mRNA levels of both PlGF and AngII were significantly increased compared to moderately stimulated cells (PlGF: (−): 1.18 ± 0.19, Aβ: 3.28 ± 0.64, *p* < 0.05; AngII: (−): 1.08 ± 0.08, Aβ: 4.68 ± 1.8 *p* < 0.05) (Figure 6A). Evaluation of AngII expression at protein levels by immunocytochemistry also confirmed the Aβ_1–42_-induced increased expression of AngII in NMW7 culture (Figure 6B and Appendix A).

## 3. Discussion

In this study, we demonstrated that a pathological angiogenesis process is active in AD model mouse brains from an early age, which is not dependent on vascular dysfunction or hypoxic condition. Moreover, we elucidated the underlying molecular mechanism of such an early angiogenesis process, where PlGF and AngII play an important role. Since this is a transgenic model that expresses an increased amount of APP, the probable cause of such angiogenesis could be increased levels of Aβ peptide. Further in vitro experiments confirmed that Aβ indeed has the ability to increase PlGF and AngII expression in neuron cultures. Angiogenesis, especially vascular dysfunction, and subsequent hypoxia-dependent pathological angiogenesis are reported in the brains of AD subjects, which is manifested as a redistribution of tight junction proteins and impaired blood-brain barrier function [27,28,29]. Such angiogenesis type and vascular dysfunction are suggested to play an important role in the development and progression of AD [14,17]. Hence, understanding the molecular mechanism of this process could be important for developing a new therapeutic intervention that targets angiogenesis and BBB restoration. Such therapy could not only inhibit the onset, but also slow the progression of AD pathology. In this respect, the findings of our study are important because we demonstrated that in addition to vascular dysfunction-mediated hypoxic angiogenesis, Aβ-dependent angiogenesis exists in the AD condition, and it appeared early in the pathology. Such findings may help to devise a more effective strategy to combat angiogenesis and thereby the onset and progression of AD pathology.

Staining data of the vessels, especially endoglin-positive new vessels, showed that the numbers were increased in the cortical areas of APP transgenic mice, whereas in the hippocampal region, they were largely unaffected. These results suggest that angiogenesis starts mainly in the cortical region of the mice at early time points, which may then spread to hippocampal areas. Aβ positive neurons are widely spread in the cortical areas, whereas the positive cells were found in compact areas at the pyramidal cell layer and dentate gyrus of the hippocampus. Such distribution might initially increase the Aβ-induced expression of angiogenesis regulators in a wide area of the cortex. Indeed, angiogenesis regulators, including PlGF and AngII, were mainly increased in the cortex, emitting a strong signal that induces angiogenesis in these areas. In humans, cerebral amyloid angiopathy and related vascular dysfunction are suggested to affect small vessels in the cortical areas [30,31]. Moreover, amyloid deposits start in the cortical areas and spread to the hippocampal areas at a later stage [32,33]. These findings suggest that cortical areas are the initial target of Aβ-dependent vascular pathology and hypoxia-dependent angiogenesis. In this report, we demonstrated that PlGF-mediated angiogenesis signals exist in the same areas early in the disease process before the development of hypoxic conditions or vessel amyloid deposits. Another important aspect of this type of angiogenesis is that AngII levels were increased without affecting AngI levels. The balance and synchronized expression of AngI and AngII is necessary for effective angiogenesis, because AngI is known to stabilize newly formed vessels, and AngII antagonizes this effect [34,35,36]. Consequently, increased expression of AngII might prevent the stabilization of newly formed vessels, resulting in pathological angiogenesis. In addition, VEGF family proteins, including PlGF, are known to induce angiogenesis by destabilizing the vessels and reducing endothelial tight junction proteins [37]. Hence, the combined effects of increased PlGF and AngII might induce pathological angiogenesis at this early time point.

Decreased basement proteins and endothelial tight junction complexes are considered markers of pathological angiogenesis [38,39]. Here, we showed that some of the newly formed vessels are devoid of basement membrane protein collagen4. Moreover, in a recent report, we have shown that tight junction protein claudin-5 levels are decreased in this AD model mouse at 2 months [40], indicating that here, angiogenesis is a pathological type. Although the cause of decreased tight junction protein claudin-5 in such pathological conditions could be due to increased PlGF protein, the reduction of collagen4 requires some protease activity. In angiogenesis conditions, proteases, including matrix metalloprotease 9 (MMP9), are considered important [41,42]. MMP9 has been shown to be increased in AD [43]. Such increased MMPs might participate in the angiogenesis process by degrading matrix proteins and tight junction complexes, along with other angiogenesis regulators. It will be interesting to investigate the regulations of protease activities at earlier time points in AD models, along with their relationships with the pathological processes.

Several reports of both animal models and human post-mortem studies demonstrated the presence of pathological angiogenesis in AD, which is suggested to be the consequence of impaired cerebral blood flow seen in AD [27,44,45]. The cause of impaired blood flow could be due to Aβ deposition and subsequent pathological changes in cerebral blood vessels [46]. In response, the expression of hypoxia-inducing factor 1α (HIF-1α) and its downstream factors, including VEGF expression, are increased, leading to a pathological angiogenic condition [47]. In our model of AD, we find that Aβ deposition around cerebral vessels is not extensive at 2 months, at which time they mainly showed an intracellular localization in the neuron-like cells. Moreover, HIF-1α protein levels were low at this time point. These results suggest that at an early time point, HIF-1α- and VEGF-dependent angiogenesis might not be important. However, Aβ peptide is known to induce an inflammatory condition, such as the expression of IL-1β, that may induce VEGF expression [48,49]. Since the neuroinflammatory condition is found to increase with time in this mouse model, such neuroinflammation-induced angiogenesis might also be important, and it should be investigated in this model in a time-dependent manner.

In vessel analysis experiments, we observed that both the total vessel numbers and the endoglin-positive new vessel numbers were increased in the APP transgenic mice cortex at 2 months of age. However, the difference in endoglin-positive new vessel numbers between APP transgenic mice and their wild-type counterparts was more pronounced than the difference in total vessel numbers. Such differences in total and new vessel numbers might be caused by the simultaneous presence of angiogenesis and vessel degradation signals in this area. The Aβ peptide showed a direct inhibitory effect on endothelial cell proliferation, and it induces apoptosis [15,27,50]. Hence, endothelial cell death by Aβ might have a negative effect on the difference in total vessel numbers between APP transgenic mice and their wild-type counterparts.

As a source of PlGF and AngII, we found that neurons can produce both, especially in the cortical areas. PlGF was found to be almost exclusively expressed by neurons, whereas AngII-positive neurons were very few. The morphology of the majority of AngII-positive cells was round-shaped, indicating the microglial type. Although we did not evaluate the involvement of microglia, our in vitro neuronal culture study demonstrated that the Aβ peptide can directly increase the mRNA expression of both PlGF and AngII in the neurons. Previous studies showed that both AngII and PlGF expression can be regulated by NF-κB transcription factors [51,52]. In fact, NOX2-mediated ROS production is important for NF-κB activation and subsequent AngII expression [52]. In neurons, Aβ has the ability to increase NOX2 activity and ROS production [53]. Additionally, ROS can activate NF-κB in neurons [53]. Taken together, it is possible that Aβ-induced ROS production activates NF-κB in neurons, which leads to the induction of PlGF and AngII. PlGF can also be regulated by endoplasmic reticulum (ER) stress and inflammation [54,55]. Aβ can cause ER stress in the neurons and neuroinflammation [56]. Moreover, our immunostaining results showed that intracellular Aβ was oligomerized. Such oligomerized Aβ might regulate the ER stress and neuroinflammation in a way that affects the expression of PlGF. Nevertheless, a detailed study is necessary to understand the exact mechanisms of how Aβ regulates PlGF expression.

## 4. Materials and Methods

### 4.1. Animals and Brain Tissue Preparation

In this study, B6.Cg-Zbtb20^Tg(PDGFB-APPSwInd)20Lms^/2Mmjax mice, commonly known as J20, were used as an AD model. Both J20 and their wild-type littermates were generous gifts from Dr. Makoto Michikawa of Nagoya City University, Japan. This transgenic mouse model expresses human amyloid precursor proteins harboring both the Swedish (K670N/M671L) and the Indiana (V717F) mutations. As a control, a non-transgenic littermate of the same age was used. All animal experimental procedures were approved by the ethical committee of Shimane University, and the animals were handled according to the guidelines of the Animal Institute of Shimane University and the guidelines of the Declaration of Helsinki. Animals were kept under a constant room temperature of 23 ± 2 °C under a 12 h light-dark cycle, with free access to water and normal chow. For immunohistochemical analysis, both J20 transgenic mice and their wild-type littermates at 2 months and 15 months (5 mice in a group) of age were deeply anesthetized with isoflurane and transcardially perfused with normal saline and 4% paraformaldehyde. The brains were extracted, postfixed, and cryoprotected, and 2 mm thick tissue blocks were prepared.

### 4.2. Immunohistochemical Analysis and Quantitation

For staining, 8 μm thick tissue slices were sectioned on a cryostat (Leica biosystem, Buffalo Grove, IL, USA). Tissue sections were treated with a blocking solution (5% normal goat or horse serum, 0.2% Triton X-100 in PBS) for 30 min, followed by incubation in anti-Aβ IgG (6E10, Rabbit, 1:200, Novus, Continental, CO, USA), anti-CD105 IgG (rat, 1:200, BioLegend, San Diego, CA, USA), anti-collagen4 IgG (rabbit, 1:200, Abcam, Cambridge, UK), anti-VEGF IgG (rabbit, 1:200, Santa Cruz, Dallas, TX, USA), anti-HIF-1α IgG (mouse, 1:200, Santa Cruz, CA, USA), anti-PlGF IgG (rabbit, 1:100, ProteinTech, Chicago, IL, USA), anti-APP IgG (rabbit, 1:100, AnaSpec, San Jose, CA, USA), anti-NeuN IgG (mouse, 1:200, Millipore), anti-oligomer IgG (A11, rabbit, 1:50, Invitrogen, Carlsbad CA, USA), or anti-AngII IgG (rabbit, 1:100, Novus) overnight at 4 °C. For the detection of immunoreactive proteins with fluorophores, the tissue sections were treated for 1 h at room temperature with species-specific IgG conjugated with Texas Red or FITC. During light microscopy, the section was treated with species-specific IgG conjugated with biotin (1:100, Vector, Ingold Road, CA, USA) at room temperature for 1 h. Then the tissue was treated with an avidin-biotin-peroxidase complex (ABC, Vector, Burlingame, CA, USA) for 30 min at room temperature. The immune reaction products were visualized with 3, 30-diaminobenzidine (DAB, Sigma, St. Louis, MO, USA) and counterstained with hematoxylin. Stained sections were examined under a fluorescent microscope (NIKON, ECLIPSE E600). Two tissue sections about 1 mm apart, starting from −1.54 mm from bregma to −2.7 mm, were used for the quantification of immunoreactive areas in the hippocampus. For the frontal cortex, two tissue sections of about 1 mm apart, starting from +0.5 mm to −0.5 mm from bregma, were used. Photomicrographs were taken at ×400 magnifications in five random microscopic fields of the designated areas. The immunoreactive areas were evaluated using ImageJ and expressed as a percent of the total area of the field. When immune reactions were detected by DAB, the IHC profiler Plugins of ImageJ were used for the quantification of the areas.

### 4.3. Solanum Tuberosum Lectin (STL) Staining

To identify vessels in the brain tissues, FITC-conjugated STL was used. After a brief wash with PBS, an 8 μm thick brain tissue section was incubated with STL (1:200, Vector) for 1 h. The tissue was washed 3 times for 5 min with PBS, mounted with a water-based mount medium, and examined under a fluorescent microscope (NIKON, ECLIPSE E600). Photomicrographs were taken at ×400 magnifications in five random microscopic fields of the designated areas, and the vessels were counted using ImageJ.

### 4.4. Cell Culture

A neural stem cell (NSC) line (NMW7) was generated from a mouse fetal brain, as described previously [57]. The cells were cultured with medium containing high glucose DMEM (Wako Pure Chemicals, Richmond, VA, USA): F12 ham (Wako) 1:1, bFGF (PeproTech, Rocky Hill, NJ, USA), 20 ng/mL, EGF (peproTech), 20 ng/mL, N2 supplement (ThermoFisher, Waltham, MA, USA), and 2% FBS (Gibco, Invitrogen) in an attached culture condition. The NSC was sub-cultured every 48 h. During stimulation, high glucose DMEM medium containing 0.2% FBS, with or without indicated concentrations of Aβ_1–42_ (Peptide Institute, Osaka, Japan), was used. Aβ_1–42_ was added to the culture as a monomer, and the stimulations were continued for the indicated times.

### 4.5. Total RNA Isolation, Reverse Transcription, and Quantitative Real-Time PCR

Total RNA was isolated from cultured cells after appropriate treatment, or from the cortical or hippocampal tissues of the mice using Trizol reagent (Invitrogen), according to the manufacturer’s instructions. To prepare the first strand cDNA, 2 μg of total RNA was reverse transcribed with reverse transcriptase enzyme (RiverTraAce, Toyobo, Osaka, Japan) in a 20 μL reaction mixture. To analyze mRNA levels, real-time PCR was performed with a SyBr green PCR system (Applied Biosystem, Warrington, UK) and appropriate gene-specific primers using an ABI Prism 7800 Sequence Detector system (Applied Biosystems). The mRNA level was normalized by corresponding GAPDH mRNA and quantified using the relative quantification method.

### 4.6. Immunocytochemistry

For immunocytochemistry, NMW7 cells were cultured in the wells of 8-well chamber slides. After appropriate treatment, the cells were fixed with 4% paraformaldehyde in PBS for 10 min. Cells were incubated in a blocking solution (5% normal goat serum, 0.5% TritonX100 in PBS) for 30 min and then incubated with anti-AngII IgG (Novus) overnight at 4 °C. The cells were treated with goat anti-rabbit IgG conjugated with biotin (1:100, Vector) at room temperature for 1 h. Then, the tissue was treated with an avidin-biotin-peroxidase complex (ABC, Vector) for 30 min at room temperature. The immune reaction products were visualized with 3, 30-diaminobenzidine (DAB, Sigma, St. Louis, MO, USA) and counterstained with hematoxylin. For fluorescent microscopy, the immunoreactive protein was detected using FITC-conjugated goat anti-rabbit IgG (1:100, Santa Cruz), and the fluorescence signals were examined under a fluorescent microscope (NIKON, ECLIPSE E600). Nuclei were identified with Hoechst. The fluorescent intensities were quantified using ImageJ.

### 4.7. Statistical Analysis

All numerical data are presented here as average ± standard deviation (SD). The statistical analysis to evaluate the differences between the two groups was performed using T TEST (Microsoft Excel).

## 5. Conclusions

In conclusion, our result demonstrated that a pathological angiogenesis process and the levels of angiogenesis regulators, including PlGF and AngII, were increased in an Alzheimer’s disease mouse model at an earlier time when HIF-1α expression was not changed. Such increased levels of angiogenesis regulators could be important for the pathology of Alzheimer’s disease.

## Figures and Tables

**Figure 1 ijms-24-04510-f001:**
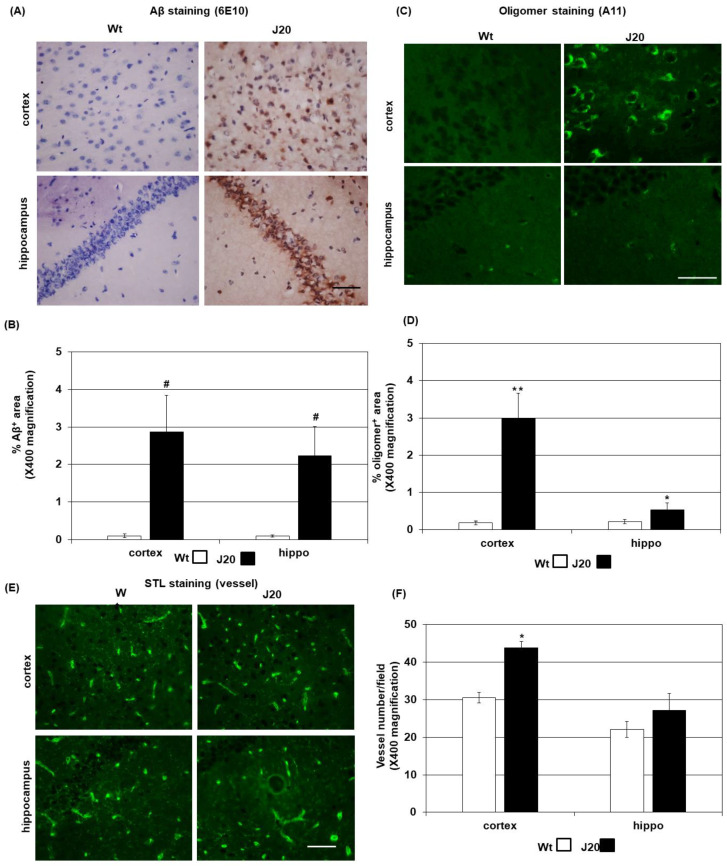
Aβ deposition and vessel density in J20 mice brain. Representative Aβ immunostaining photomicrographs of the prefrontal cortex and hippocampal areas are shown in (**A**); the brown areas are immunopositive; the hematoxylin counterstained nuclei are blue. The immunopositive area in a field at ×400 magnification was quantified using ImageJ, and the average quantified data (*n* = 5 in each group) are presented in (**B**). Oligomerization of the proteins was assessed by immunostaining using an oligomer-specific antibody. FITC-conjugated species-specific secondary antibody was used to detect the immunoreactive oligomers, which is depicted here as green areas. Representative oligomer immunostaining photomicrographs of the prefrontal cortex and hippocampal areas are shown in (**C**). Anti-oligomer immunopositive areas were quantified, and the average quantified data (*n* = 5 in each group) are presented in (**D**). Vessels were identified with STL staining. Representative photomicrographs of vessels in the prefrontal cortex and hippocampal areas are shown in (**E**), where the green areas indicate vessels. Vessels were counted in 5 random microscopic fields at ×400 magnification, as described in the Materials and Methods section, and are presented in (**F**). The numerical data are presented here as averages ± SD (*n* = 5). Statistical significance is denoted as follows: * *p* < 0.05, ** *p* < 0.01, ^#^
*p* < 0.001 vs. wild-type (Wt) mice. Bar = 50 µM.

**Figure 2 ijms-24-04510-f002:**
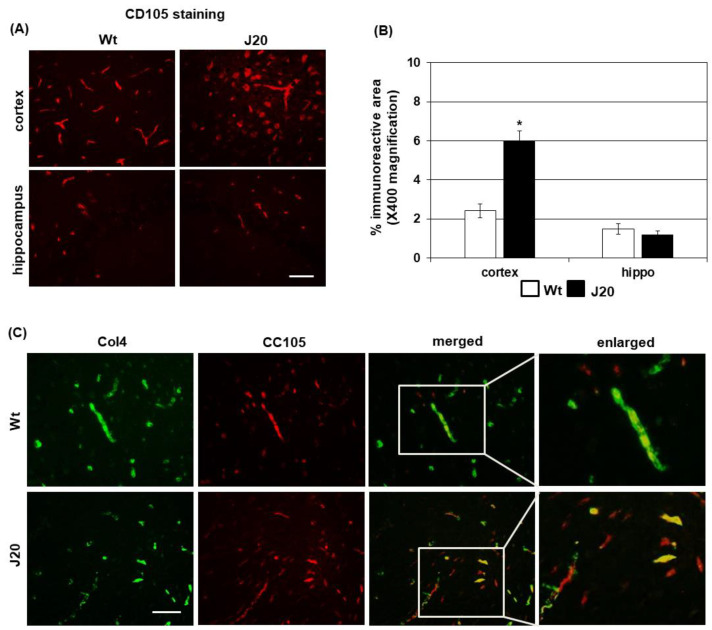
Angiogenesis in J20 mice brains. Immunostaining of CD105 was used to identify newly formed vessels. Representative photomicrographs of CD105-positive vessels (red) in the prefrontal cortex and hippocampal areas are shown in (**A**). (**B**) CD105-positive vessels were quantified in 5 random microscopic fields at ×400 magnification and are presented as average ± SD (*n* = 5) of % immunopositive area. To assess the angiogenesis type, double immunofluorescence staining of CD105 and collagen4 (Col4) was performed. Representative photomicrographs of CD105 (red) and Col4 (green) and their merged pictures are shown in (**C**). Statistical significances are denoted as follows: * *p* < 0.05 vs. wild-type (Wt) mice. Bar = 50 µM.

**Figure 3 ijms-24-04510-f003:**
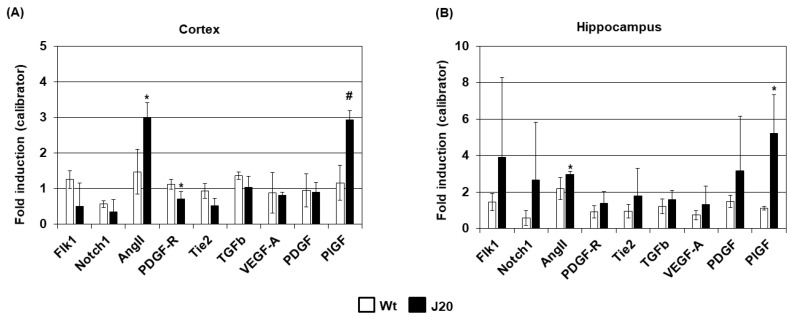
Expression of angiogenesis regulators at mRNA levels in J20 mice brains. For quantification, the cDNA of the target mRNA and the GAPDH mRNA were PCR amplified simultaneously. The GAPDH-normalized mRNA levels of the target genes were expressed as the fold induction relative to a calibrator, where the sample of a wild-type mouse served as such. The real-time PCR results of the prefrontal cortex are shown in (**A**), and those of the hippocampus are shown in (**B**). The numerical data are presented here as averages ± SD (*n* = 5). Statistical significance is denoted as follows: * *p* < 0.05 vs. wild-type (Wt) mice; # *p* < 0.01 vs. wild-type (Wt) mice.

**Figure 4 ijms-24-04510-f004:**
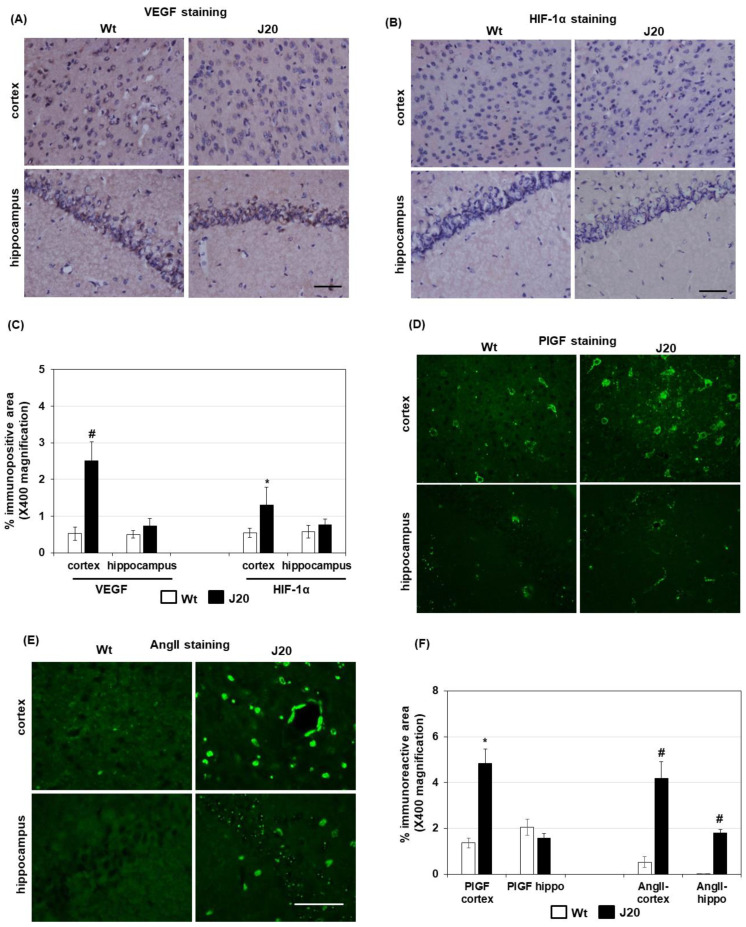
Expression of angiogenesis regulators at protein levels in J20 mice brains. Angiogenesis regulators including VEGF, HIF-1α, PlGF, and AngII were evaluated by immunostaining. Representative immunostaining photomicrographs of VEGF and HIF-1α in the prefrontal cortex and the hippocampal areas are shown in (**A**) and (**B**), respectively. Immunopositive areas of VEGF and HIF-1α in 5 random microscopic fields at ×400 magnification were quantified using ImageJ, and presented in (**C**). Representative immunofluorescence staining photomicrographs of PlGF and AngII in the prefrontal cortex and the hippocampal areas are shown in (**D**) and (**E**), respectively, where the immunopositive areas are depicted in green. Immunopositive areas of PlGF and AngII in 5 random microscopic fields at ×400 magnification were quantified using ImageJ, and presented in (**F**). The numerical data are presented here as averages ± SD (*n* = 5). Statistical significance is denoted as follows: * *p* < 0.05 vs. wild-type (Wt) mice; # *p* < 0.01 vs. wild-type (Wt) mice. Bar = 50 µM.

**Figure 5 ijms-24-04510-f005:**
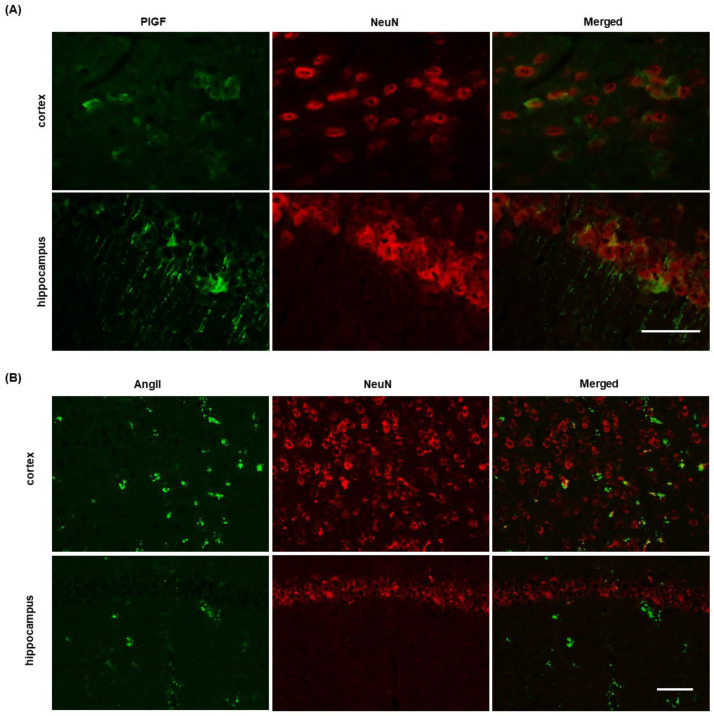
Identification of PlGF and AngII expressing cells in J20 mice brains. PlGF- and AngII-expressing cells in the brains of J20 mice were identified by double immunofluorescence staining, where NeuN was used as a neuron marker. Representative photomicrographs of PlGF (green) and NeuN (red) and their merged images are shown in (**A**). In (**B**), representative photomicrographs of AngII (green) and NeuN (red) and their merged images are shown. Bar = 50 µM.

**Figure 6 ijms-24-04510-f006:**
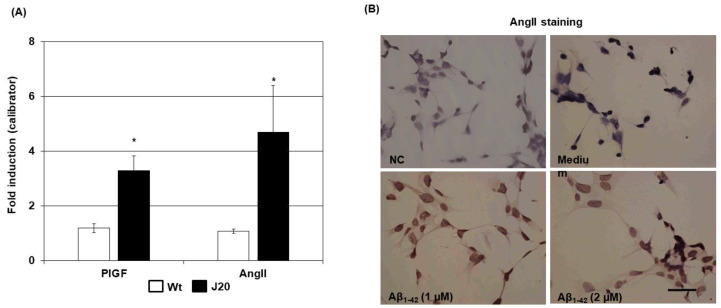
Effects of Aβ on the expression of PlGF and AngII in neuronal culture. (**A**) An NSC line (NMW7) was treated with synthetic Aβ1–_42_ peptide (2 µM) for 24 h, mRNA was isolated, and real-time PCR was performed to evaluate PlGF and AngII mRNA levels. The mRNA levels of target genes were normalized with the GAPDH mRNA of the same sample and expressed as the fold induction relative to a calibrator, where 1 sample of the medium-stimulated (−) condition served as such. The numerical data are presented here as averages ± SD (*n* = 5 experiments). Statistical significance is denoted as follows: * *p* < 0.01 vs. moderately stimulated (−) condition. (**B**) AngII was further evaluated at protein levels by immunocytochemistry after stimulating NMW7 with Aβ_1–42_ (1 and 2 µM) for 48 h. Representative photomicrographs of (−) and Aβ_1–42_ (1 and 2 µM) stimulated NMW7 are shown. An Aβ_1–42_ (2 µM) stimulated culture was immunostained, in which normal rabbit IgG was used as the primary antibody instead of anti-AngII IgG (negative control) to evaluate nonspecific staining. A representative photomicrograph of negative control staining (NC) is shown in (**B**). Bar = 50 µM.

## Data Availability

All data of this study are included in the manuscript, or in the Appendix A.

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
