# Peer review of "Alzheimer’s Amyloid β Peptide Induces Angiogenesis in an Alzheimer’s Disease Model Mouse through Placental Growth Factor and Angiopoietin 2 Expressions"

_ijms, 2023, doi:10.3390/ijms24054510_

Round 1
Reviewer 1 Report
​​​Abstract - The authors state "Angiogenesis is known to exist in Alzheimer’s disease (AD) brains, .." Please consider altering this sentence to "Increased angiogenesis has been documented in Alzheimer’s disease (AD) brains, .." or similar. - " To understand the direct effects of Alzheimer’s amyloid β (Aβ) peptide on angiogenesis, we checked the mechanism in the brain of young (2 months old) APP transgenic AD model mice (J20 strain)." In this sentence, please delete "(2 months old)" and "(J20 strain)" -- This level of detail is not needed in an abstract. Please also consider re-writing the sentence as "To understand the role of amyloid β (Aβ) peptide on angiogenesis, we analyzed effects on the brains of young APP transgenic mice." Introduction - The authors state " Aggregated Aβ shows neurodegener- 39 ative and neuroinflammatory properties." Please add a citation after this sentence. - "After production, Aβ is 43 cleared from the brain by enzymatic degradation and by phagocytic cells." Please add a citation after this sentence. - " In this study, we aimed to investigate the direct effects of Aβ on angiogenesis in AD 62 using disease model mice and in vitro cell culture systems." Please consider changing to: " In this study, we investigated the direct effects of Aβ on angiogenesis in AD using disease model mice and in vitro cell culture systems." Results - The authors state "Angiogenesis is known to exist in AD brains .." Please consider altering this sentence to "Increased angiogenesis has been documented in Alzheimer’s disease (AD) brains, .." or similar. Also, a citation is needed after this sentence. 2.1. Aβ deposition and vessel density in J20 mice brains: - If the authors could add some numerical data in this section it would strengthen the description (e.g., mean labeling values with range, p values, etc). 2.4 - The authors state: "To identify the cells that express PlGF and AngII in the J20 mouse brains, double immunofluorescence staining was done, where NeuN was used as a marker of neurons." Please re-write this sentence: "To identify whether neurons expressed PlGF and AngII in J20 mouse brains, double immunofluorescence experiments were performed using NeuN neuronal labeling." The authors should please address the following issues with regard to images: - Remove "Figure 1", "Figure 2" ... "Figure 4" labels from the figures.​ - Enlarge the text size throughout. - Black/white bar graphs may be ideal. Figure 1 caption - Please remove "Aβ deposition in the J20 mice brain 94 was evaluated by immunostaining." This sentence is not needed in the caption. Figure 2 caption - Please remove "Angiogenesis in J20 mice brain was evaluated by im- 107 munostaining, as described in the Materials and Methods." ​ This sentence is not needed in the caption. Figure 3 caption - Please remove " The expression of angiogenesis regulators at mRNA levels in J20 mice brains was evaluated by real-time PCR, as described in the Materials and Methods."​ ​For all the captions, please consider re-writing with more useful description of the nice data that is shown. No need to describe methods details in the figure captions. More descriptive captions will strengthen the paper and help the readers. For all the H&E/Immunoperoxidase images, please consider increasing the brightness.​ High-power insets of H&E/Immunoperoxidase/Immunofluorescence images (close cropped cells and/or zoom-in images) will also help. ​Methods: 4.1: Animals - The authors describe K670N/M671L and V717F mice. Were animals from Jackson Lab, or elsewhere? Please specify from where they were obtained.​ 4.2: Immunohistochemical analysis - The authors state the following: "Both J20 transgenic mice and their wild-type littermates at 2 months and 15 months (5 mice in a group) of age were deeply anesthetized with Isoflurane and transcardially perfused with normal saline and 4% paraformaldehyde. The brain was postfixed, cryoprotected and 2 mm thick tissue blocks were prepared." This sentence should please be moved to the prior section (4.1) The title of 4.1 can be changed to "Animals and Tissues" or "Animals and Brain Harvesting" Please add that the brains "were extracted" ("The brain was extracted, postfixed, and cryoprotected and then 2 mm thick tissue blocks were prepared.") - Please add the type of cryostat used for sectioning (vendor/model) - Please check for typos (e.g., "tinental, CO, USA),anti-CD105 IgG" needs a space added) - Since immunohistochemistry quantitation method is also described in this section, please consider changing the section title to "Immunohistochemical analysis and Quantitation", or similar.Author Response
First of all, we thank the reviewers for their constructive review. All of the points raised by the reviewers are very helpful. We have changed the manuscript following all of their suggestion.
Reviewer 1:
Abstract - The authors state "Angiogenesis is known to exist in Alzheimer’s disease (AD) brains, .." Please consider altering this sentence to "Increased angiogenesis has been documented in Alzheimer’s disease (AD) brains, .." or similar.
Response: According to the reviewer’s comment, we have changed the sentence.
- " To understand the direct effects of Alzheimer’s amyloid β (Aβ) peptide on angiogenesis, we checked the mechanism in the brain of young (2 months old) APP transgenic AD model mice (J20 strain)." In this sentence, please delete "(2 months old)" and "(J20 strain)" -- This level of detail is not needed in an abstract.
Response: according to the reviewer’s suggestion, we have deleted those parts of the sentence.
Please also consider re-writing the sentence as "To understand the role of amyloid β (Aβ) peptide on angiogenesis, we analyzed effects on the brains of young APP transgenic mice."
Response: According to the reviewer’s comment, the sentence has been re-written.
Introduction - The authors state " Aggregated Aβ shows neurodegenerative and neuroinflammatory properties." Please add a citation after this sentence.
Response: Here, the following 2 references are added to the revised manuscript.
Ref-1. Ismail R, Parbo P, Madsen LS, Hansen AK, Hansen KV, Schaldemose JL, Kjeldsen PL, Stokholm MG, Gottrup H, Eskildsen SF, Brooks DJ. The relationships between neuroinflammation, beta-amyloid and tau deposition in Alzheimer’s disease: a longitudinal PET study. J Neuroinflammation. 2020 May 6;17(1):151. doi: 10.1186/s12974-020-01820-6.
Ref2-
Zhu L, Li R, Jiao S, Wei J, Yan Y, Wang ZA, Li J, Du Y. Blood-Brain Barrier Permeable Chitosan Oligosaccharides Interfere with β-Amyloid Aggregation and Alleviate β-Amyloid Protein Mediated Neurotoxicity and Neuroinflammation in a Dose- and Degree of Polymerization-Dependent Manner. Mar Drugs. 2020 Sep 25;18(10):488. doi: 10.3390/md18100488.
"After production, Aβ is cleared from the brain by enzymatic degradation and by phagocytic cells." Please add a citation after this sentence.
Response: The following 4 references are added to the revised manuscript.
Ref-1
- Lee CYD, Landreth GE. The role of microglia in amyloid clearance from the AD brain. J Neural Transm. 2010; 117:949-960.
- Ries M, Sastre M. Mechanisms of Aβ clearance and degradation by glial Cells. Front Aging Neurosci. 2016; 8:160. doi: 10.3389/fnagi.2016.00160.
- Saido T, Leissring MA. Proteolytic degradation of amyloid β-protein. Cold Spring Harb Perspect Med. 2012; 2(6):a006379. doi: 10.1101/cshperspect.a006379.
- Numata K, Kaplan DL. Mechanisms of enzymatic degradation of amyloid Beta microfibrils generating nanofilaments and nanospheres related to cytotoxicity. Biochemistry. 2010; 49:3254-3260.
- " In this study, we aimed to investigate the direct effects of Aβ on angiogenesis in AD using disease model mice and in vitro cell culture systems." Please consider changing to: " In this study, we investigated the direct effects of Aβ on angiogenesis in AD using disease model mice and in vitro cell culture systems."
Response: according to the reviewer’s comment, we have amended the sentence.
Results - The authors state "Angiogenesis is known to exist in AD brains .." Please consider altering this sentence to "Increased angiogenesis has been documented in Alzheimer’s disease (AD) brains, .." or similar. Also, a citation is needed after this sentence.
Response: according to the reviewer’s comment we amended the sentence and added 2 references
Ref1. Vagnucci Jr AH, Li WW Alzheimer's disease and angiogenesis. Lancet 2003; 361(9357):605-608.
Ref2. Grammas P, Tripathy D, Sanchez A, Yin X, Luo J. Brain microvasculature and hypoxia-related proteins in Alzheimer's disease. Int J Clin Exp Pathol. 2011; 4:616-627.
2.1. Aβ deposition and vessel density in J20 mice brains: - If the authors could add some numerical data in this section it would strengthen the description (e.g., mean labeling values with range, p values, etc).
Response: according to the reviewer’s comment, we have included the numerical data in the text.
2.4 - The authors state: "To identify the cells that express PlGF and AngII in the J20 mouse brains, double immunofluorescence staining was done, where NeuN was used as a marker of neurons." Please re-write this sentence: "To identify whether neurons expressed PlGF and AngII in J20 mouse brains, double immunofluorescence experiments were performed using NeuN neuronal labeling."
Response: we have amended the sentence according to the reviewer’s suggestion.
The authors should please address the following issues with regard to images: - Remove "Figure 1", "Figure 2" ... "Figure 4" labels from the figures.​- Enlarge the text size throughout. - Black/white bar graphs may be ideal.
Response: According to the reviewer’s suggestion we have amended the figures.
Figure 1 caption - Please remove "Aβ deposition in the J20 mice brain 94 was evaluated by immunostaining." This sentence is not needed in the caption.
Response: We have removed the sentence according to the reviewer’s suggestion.
Figure 2 caption - Please remove "Angiogenesis in J20 mice brain was evaluated by immunostaining, as described in the Materials and Methods."​ This sentence is not needed in the caption.
Response: We have removed the sentence according to the reviewer’s suggestion.
Figure 3 caption - Please remove " The expression of angiogenesis regulators at mRNA levels in J20 mice brains was evaluated by real-time PCR, as described in the Materials and Methods."​​For all the captions, please consider re-writing with more useful description of the nice data that is shown. No need to describe methods details in the figure captions. More descriptive captions will strengthen the paper and help the readers.
Response: We have removed the sentence according to the reviewer’s suggestion.
For all the H&E/Immunoperoxidase images, please consider increasing the brightness.​High-power insets of H&E/Immunoperoxidase/Immunofluorescence images (close cropped cells and/or zoom-in images) will also help. ​
Response: According to the reviewer’s suggestion, we have increased the brightness of all Immunoperoxidase images.
Methods: 4.1: Animals - The authors describe K670N/M671L and V717F mice. Were animals from Jackson Lab, or elsewhere? Please specify from where they were obtained.​
Response: This information is added in the revised manuscript.
4.2: Immunohistochemical analysis - The authors state the following: "Both J20 transgenic mice and their wild-type littermates at 2 months and 15 months (5 mice in a group) of age were deeply anesthetized with Isoflurane and transcardially perfused with normal saline and 4% paraformaldehyde. The brain was postfixed, cryoprotected and 2 mm thick tissue blocks were prepared." This sentence should please be moved to the prior section (4.1) The title of 4.1 can be changed to "Animals and Tissues" or "Animals and Brain Harvesting" Please add that the brains "were extracted" ("The brain was extracted, postfixed, and cryoprotected and then 2 mm thick tissue blocks were prepared.")
Response: We have amended the text according to the reviewer’s suggestion.
- Please add the type of cryostat used for sectioning (vendor/model) - Please check for typos (e.g., "tinental, CO, USA),anti-CD105 IgG" needs a space added) –
Response: This information is added to the revised manuscript.
Since immunohistochemistry quantitation method is also described in this section, please consider changing the section title to "Immunohistochemical analysis and Quantitation", or similar.
Response: According to the reviewer’s comment the part is amended.。
Reviewer 2 Report
In present work, Sheikh et al. have investigated the possible involvement of AB of AD in Angiogenesis seen in the pathology of AD. I this regard they evaluated the cortex and hippocampus of transgenic mice model of AD and have shown that reconfirmed the phenomena of angiogenesis and also have shown that two angiogenic factors, placental growth factor 3
and angiopoietin 2, are also overexpressed in these brain regions. In their study they also have shown that AB is able to induce these two angiogenic factors in an in-vitro study, concluding that the AB is responsible for Angiogenesis seen in the pathology of AD. In my view, this is an interesting subject, and less focused aspects of AD pathology have been discussed.There are, however, some points that need to be resolved before taking a decision to publish in IJMS.
The first major point is that most of the results are not novel. The study has shown overexpression of Aβ and markers of vascularization, which are referenced in previous studies. On the other hand, as mentioned in the introduction section "It is suggested that Aβ deposition might cause dysfunction of vessel function resulting in a hypoxic condition in AD brains, which triggers the angiogenic process" therefore the relation between Aβ and angiogenic process was also reported before. As a result, the authors should define the gaps in this field more clearly in the introduction. What is known and what is not. It is inferred from the context that authors aimed to reveal if Aβ is also able to induce angiogenesis in a direct way or not and if this is true what are the mechanisms? If this is the gap, then authors should describe it and be focussed on this part instead of repeating other known facts (such as Aβ deposition in AD and the fact of pathological vascularization in this disease).
This is also true for the discussion part, the angiogenic factors and their roles in pathological angiogenesis, and also its relation with the Aβ in this paradigm is not well covered in the discussion. Furthermore, the authors concluded that "Such increased levels of angiogenesis regulators could be important for the pathology of Alzheimer’s disease". Why this could be important and how it can help us in the treatment and prevention of this disease. why authors did not discuss the possibility of antagonizing these factors and do not bring data about these factors....
Author Response
First of all, we thank the reviewers for their constructive review. All of the points raised by the reviewers are very helpful. We have changed the manuscript following all of their suggestion.
Reviewer 2:
In present work, Sheikh et al. have investigated the possible involvement of AB of AD in Angiogenesis seen in the pathology of AD. I this regard they evaluated the cortex and hippocampus of transgenic mice model of AD and have shown that reconfirmed the phenomena of angiogenesis and also have shown that two angiogenic factors, placental growth factor and angiopoietin 2, are also overexpressed in these brain regions. In their study they also have shown that AB is able to induce these two angiogenic factors in an in-vitro study, concluding that the AB is responsible for Angiogenesis seen in the pathology of AD. In my view, this is an interesting subject, and less focused aspects of AD pathology have been discussed. There are, however, some points that need to be resolved before taking a decision to publish in IJMS.
The first major point is that most of the results are not novel. The study has shown overexpression of Aβ and markers of vascularization, which are referenced in previous studies. On the other hand, as mentioned in the introduction section "It is suggested that Aβ deposition might cause dysfunction of vessel function resulting in a hypoxic condition in AD brains, which triggers the angiogenic process" therefore the relation between Aβ and angiogenic process was also reported before. As a result, the authors should define the gaps in this field more clearly in the introduction. What is known and what is not. It is inferred from the context that authors aimed to reveal if Aβ is also able to induce angiogenesis in a direct way or not and if this is true what are the mechanisms? If this is the gap, then authors should describe it and be focused on this part instead of repeating other known facts (such as Aβ deposition in AD and the fact of pathological vascularization in this disease).
Response: We thank the reviewer for the comment. Indeed, some of the results are not new. For example, increased expression of Aβ in this mouse model is not new. Still, we analyzed Aβ deposition in this model to see the status of the deposited Aβ at 2 months of age, and its direct effects on angiogenesis. The reviewer correctly pointed out that previous papers showed that a hypoxic condition is the cause of angiogenesis. Here, we have investigated the direct effects of Aβ on angiogenesis. To eliminate the hypoxia factor in angiogenesis, we used young mice, where Aβ did not deposit in the vessels. And we showed that angiogenesis factors PlGF and AngII are directly regulated by Aβ. This information is given in the introduction. However, we think that the introduction was not well-written, as the reviewer pointed out. So, we amended the introduction section according to the reviewer’s suggestion.
This is also true for the discussion part, the angiogenic factors and their roles in pathological angiogenesis, and also its relation with the Aβ in this paradigm is not well covered in the discussion. Furthermore, the authors concluded that "Such increased levels of angiogenesis regulators could be important for the pathology of Alzheimer’s disease". Why this could be important and how it can help us in the treatment and prevention of this disease. why authors did not discuss the possibility of antagonizing these factors and do not bring data about these factors....
Response: We also think that the discussion part was not well-written. So, we rewrote the discussion part following the points of the reviewer. After the amendment, we think that the discussion section of the paper is vastly improved.
In the results section, some sentences are not directly representing the results of the current study, and mainly describe the rationale of the study… it is better to bring these to the introduction and/or discussion section.
Response: We understand that some of the sentences in the result sections are not directly representing results, sometimes such sentences are necessary to describe the rationale of the results, as the reviewer pointed out. However, similar information is given in the introduction section also to make the text smooth.
Also, some of the results are ambiguously written and it is needed to read other parts to understand the results, for example, in this sentence “ The staining demonstrated a wide distribution pattern in the cortex, but in the hippocampus CA1 areas, that was mainly positive in the hippocampal pyramidal cell layer (Figure 1A).” staining with which marker is pointed, is there any statistical analysis?? What are the statistical results (F, p..)? ...... This is also true for almost all data, the data is just described without any detail about the statistical details, the type of test, and the level of significance…
Response: We have amended the section according to the reviewer’s suggestion. Statistical results are also given here.
Round 2
Reviewer 1 Report
Abstract:
-- "Thus, our results suggest that pathological angiogenesis exists in AD brains due to the direct effects of Aβ from an earlier time point. Aβ regulates such angiogenesis through PlGF and AngII expression."
Recommend combining and rewording these, "Thus, these pilot data suggest that pathological angiogenesis exists in AD brains due to the direct effects of early Aβ accumulation and suggests that Aβ peptide regulates angiogenesis through PlGF and AngII expression."
Introduction:
-- "These features (neurodegeneration and neuroinflammation) are always found in AD brains, indicating the important role of aggregation and deposition of the peptide in the pathology [3]."
Request changing this to "These features (neurodegeneration and neuroinflammation) are always found in AD brains, indicating the potential importance of aggregation and deposition of the Aβ peptide in pathology [3]."
-- "Such deposition causes the vessel fragile, resulting in microbleeds and associated inflammation [13]"
Request changing this to "Such deposition causes the vessel to become fragile, resulting in microbleeds and associated inflammation [13]"
Figures:
Some of the IHC results are still somewhat difficult to discern (e.g., Fig 4E).
Author Response
We thank the reviewer for checking the manuscript meticulously. We have changed the manuscript following all suggestions of the reviewer. Now, we think that the paper improved vastly.
Abstract:
-- "Thus, our results suggest that pathological angiogenesis exists in AD brains due to the direct effects of Aβ from an earlier time point. Aβ regulates such angiogenesis through PlGF and AngII expression."
Recommend combining and rewording these, "Thus, these pilot data suggest that pathological angiogenesis exists in AD brains due to the direct effects of early Aβ accumulation and suggests that Aβ peptide regulates angiogenesis through PlGF and AngII expression."
Response: According to the reviewer’s suggestion, we have amended the sentences.
Introduction:
-- "These features (neurodegeneration and neuroinflammation) are always found in AD brains, indicating the important role of aggregation and deposition of the peptide in the pathology [3]."
Request changing this to "These features (neurodegeneration and neuroinflammation) are always found in AD brains, indicating the potential importance of aggregation and deposition of the Aβ peptide in pathology [3]."
Response: According to the reviewer’s suggestion, we have amended the sentence.
-- "Such deposition causes the vessel fragile, resulting in microbleeds and associated inflammation [13]"
Request changing this to "Such deposition causes the vessel to become fragile, resulting in microbleeds and associated inflammation [13]"
Response: According to the reviewer’s suggestion, we have amended the sentence.
Figures:
Some of the IHC results are still somewhat difficult to discern (e.g., Fig 4E).
Response: According to the reviewer’s suggestion, we have improved the figure quality.
Reviewer 2 Report
The authors have addressed my concerns, and I recommend publication in the journal.
Author Response
The authors have addressed my concerns, and I recommend publication in the journal.
Response: We thank the reviewer for the effort to improve the quality of our paper by showing us the mistakes, and with constructive criticism. We think amending the paper following those points makes our paper more understandable.